# Research on Trimming Frequency-Increasing Technology for Quartz Crystal Resonator Using Laser Etching

**DOI:** 10.3390/mi12080894

**Published:** 2021-07-28

**Authors:** Jun-Lin Zhang, Shuang Liao, Cheng Chen, Xiu-Tao Yang, Sheng-Ao Lin, Feng Tan, Bing Li, Wen-Wu Wang, Zheng-Xiang Zhong, Guang-Gen Zeng

**Affiliations:** 1College of Materials Science and Engineering, Sichuan University, Chengdu 610064, China; zhangjl.kong.di@gmail.com (J.-L.Z.); yangxiutao532624@foxmail.com (X.-T.Y.); linshengao@163.com (S.-A.L.); libing70@126.com (B.L.); www1492@163.com (W.-W.W.); 2School of Automation Engineering, University of Electronic Science and Technology of China, Chengdu 611731, China; liaoshuang1001@foxmail.com (S.L.); chencheng_UESTC@163.com (C.C.); tanfeng@uestc.edu.cn (F.T.); 3CQC-Trusted Testing Technology Co., Ltd., Nanjing 210023, China; zhongzhengxiang@cqctt.com.cn

**Keywords:** quartz crystal resonator (QCR), laser etching, trimming frequency-increasing technology

## Abstract

A quartz crystal resonator (QCR) is an indispensable electronic component in the field of the modern electronics industry. By designing and depositing electrodes of different shapes and thicknesses on a quartz wafer with a certain fundamental frequency, the desired target frequency can be obtained. Affected by factors such as the deposition equipment, mask, wafer size and placement position, it is difficult to accurately obtain the target frequency at a given time, especially for mass-produced QCRs. In this work, a laser with a wavelength of 532 nm was used to thin the electrodes of a QCR with a fundamental frequency of 10 MHz. The electrode surface was etched through a preset processing pattern to form a processing method of local thinning of the electrode surface. At the same time, the effect of laser etching on silicon dioxide and resonator performance was analyzed. Satisfactory trimming frequency-increasing results were achieved, such as a frequency modulation accuracy of 1 ppm, frequency distribution with good consistency and equivalent parameters with small changes, by the laser partial etching of the resonator electrode. However, when the surface electrode was etched into using through-holes, the attenuation amplitude of the equivalent parameter became larger, especially in terms of the quality factor (Q), which decreased from 63 K to 1 K, and some resonators which had a serious frequency drift of >40%. In this case, a certain number of QCRs were no longer excited to vibrate, which was due to the disappearance of the piezoelectric effect caused by the local thermal phase change in the quartz wafer.

## 1. Introduction

Quartz crystal resonators (QCR), which have the advantages of a stable frequency, relatively large bandwidth and a high quality factor, are among the key frequency electronic components in the modern information industry and have been widely used in various pieces of communications and electronic equipment [1,2]. The basic requirement for a QCR is to have a stable (aging < 10^−2^ ppm/day) and accurate (<5 ppm, 25 °C) frequency. As such, related research on quartz crystal frequency trimming technology is receiving increasing attention. At present, numerous studies have been conducted on the correlation between frequency and oscillation states with quartz wafer size and electrode thickness [3,4]. The classic Mindlin theory shows that the relationship between the resonance frequency of a resonator and the electrode size on its surface is extremely sensitive [5]. Depositing electrodes on quartz will increase the load of the quartz wafer and decrease the frequency. Affected by the internal structure of the deposition equipment, the thickness of the mask plate, processing accuracy, quartz wafer size and the placement position, it is difficult to obtain the QCR with a target frequency accurately at a given time. In terms of mass production, it is especially difficult to obtain QCRs with a consistent frequency. Therefore, it is very important to trim the resonance frequency of QCRs before encapsulation.

The trimming methods includes physical methods and chemical methods. According to the change in the frequency, the trimming methods can be divided into frequency-increasing technology and frequency-reducing technology. Frequency reduction can be achieved by increasing the load of the electrodes using physical or chemical deposition, such as vacuum evaporation, magnetron sputtering and electroplating. In order to trim the frequency-increasing of QCRs, ion beam etching, laser etching and chemical corrosion are used to reduce the mass of the electrode or wafer. Ion beam etching, which became popular in 1999, is an effective method to trim the frequency of quartz [6,7,8]. Ion beam etching generally does not damage the electrodes; is less affected by the mask; and has high frequency modulation accuracy, which can improve the resonator yield. However, ion beam etching will cause frequency drift in the early stage of frequency change. In addition, the equipment is huge and inflexible, with a complex etching process. The surface of the quartz wafer is easily damaged during ion beam reprocessing, especially when this is conducted at a later stage [9]. Therefore, some other frequency-increasing technologies have been developed. Raicheva et al. studied the technology of chemically etching quartz wafers using an (NH_4_)_2_F_2_ aqueous solution and revealed the optimum etching temperature. In this technique, the frequency is modulated by controlling the etching time [10]. However, if QCRs are chemically etched, the electrode thickness and dimensions are likely to change at the same time, especially at the edge of the electrode which causes unpredictable corrosion. This creates more uncertainties, such as producing additional equivalent resistance and a change in equivalent capacitance. Thus, finding better physical technology to increase frequency is the direction that researchers have been working towards.

In the modern industry, metals can be processed with precision by lasers. In 1999, Wellershoff et al. studied the energy deposition depth and transfer to the lattice for Au, Ni, and Mo films of varying thickness by femtosecond laser pulses, proving that laser damage in metals is a purely thermal process even for femtosecond pulses [11]. Hohlfeld et al. studied the hot electron diffusion in thin sheets of gold and nickel and considered the influence of valence electrons on the energy transfer process by the two-temperature model [12,13]. Venkatakrishnan et al. pointed out that clean and precise micro-structuring of gold films with a thickness of 1000 nm can be achieved with femtosecond pulsed lasers (150 fs, 400 nm, 1 kHz) by controlling the pulse energy in the first ablation regime [14]. Kamlage et al. reported the results of investigations on deep drilling of metals by femtosecond laser pulses and found that femtosecond lasers can drill deep and high-quality holes in metals without any post-processing or need for a special gas environment at high laser fluences [15]. Gamaly et al. described the mechanism of the ablation of solids by using intense femtosecond laser pulses in an explicitly analytical form and proposed the existence of long-lived non-equilibrium transition states at the solid/vacuum interface during the etching process [16,17]. Jiang et al. extended the existing two-temperature model for use with high electron temperatures by using full-run quantum treatments to calculate the significantly varying properties and found that the proposed model predicted the damage thresholds more accurately than the existing model for gold films when compared with published experimental results [18]. QCRs’ electrode materials are mainly inactive metals, such as gold, silver and platinum. According to the literature, it is feasible to trim the frequency-increasing of QCRs by laser etching. At present, there are few systematic studies in this area, and only a few scholars have attempted to trim the frequency of QCRs by etching the silver electrode with a laser [19].

Based on the above analysis, this work proposes to adopt a laser etching technique in order to trim the thickness of the gold electrode by setting a fine post-processing pattern of the electrode and changing the parameters of the laser, followed by a series of frequency-increasing studies of the QCR. The effect of laser etching on the equivalent parameters of the QCR is also analyzed in terms of the variation of the material properties of the quartz wafer, the electrode surface and the electrode/quartz interface.

## 2. Experiment

The resonance frequency of QCR has the following relationship with the frequency constant *K_f_*:(1)$$f_{n}=\frac{n}{t_{q}}K_{f}$$
where *t_q_* is the thickness of the quartz wafer; *n* is the overtone number of thickness-shear mode; and *f_n_* is the fundamental frequency when *n* = 1; *K_f_* = 1670 kHz·mm. Below is the differential of the above equation:(2)$$∆f=-\frac{K_{f}}{t_{q}^{2}}∆t_{q}$$

When an electrode is deposited on a quartz crystal, the frequency change Δ*f*_1_ in the quartz crystal is
(3)$$∆f_{1}=-\frac{K_{f}}{t_{q}^{2}}\frac{\Delta t_{m}\rho_{m}A_{2}}{\rho_{q}A_{1}}$$
where Δ*t_m_ is* the thickness of a deposited electrode, *A*_1_ is the area of the quartz wafer, *A*_2_ is the area of the electrode, *ρ_m_* is the metal electrode density, and *ρ_q_* is the density of the quartz crystal. It can be seen that the change in the resonance frequency is negatively correlated with the film thickness change. The thickness of the electrode to be deposited is calculated according to the target frequency and the above equation.

The quartz wafer used in this work is of an AT-cut type with a diameter of 8 mm and a fundamental frequency of 10 MHz, provide by SSCE Co., LTD. After a rigorous semiconductor cleaning process, the gold electrodes with a diameter of 4 mm and a thickness of 80 nm were fabricated by vacuum thermal evaporation using a mask. The resonator, with a frequency lower than the target frequency, was then mounted on the working platform in the laser frequency modulation equipment shown in Figure 1.

The etching pattern is shown in Figure 1. The relationship between the change in the QCR frequency Δ*f*_2_ and the change in the electrode thickness in the etched area can be defined by the following formula:(4)$$∆f_{2}=-\frac{K_{f}}{t_{q}^{2}}\frac{\Delta t_{m1}\rho_{m}A_{3}}{\rho_{q}A_{1}}$$
where *A*_3_ is the area of the etched pattern, and Δ*t_m1_* is the thickness change in etched area. This equation was then used to calculate the thickness of the electrode to be etched by the laser. Laser etching parameters in this work were as follows: pulse repetition frequency (PRF) = 10 kHz, *Is* = 23 A, line spacing = 0.01 mm, power factor (PWF) = 120–750, scanning speed = 5–1000 mm/s. As shown in Figure 1b–d, laser etching effects included the following three types: type a, in which the electrodes were partially etched; type b, in which the electrode on one side of the QCR was etched into through-holes; and type c, in which the electrodes on both sides were etched into through-holes.

The equivalent parameter characteristics of the QCRs were measured using the precision impedance analyzer (4294A, Agilent, CA, USA). The morphology of the electrode and silicon dioxide after etching were measured using SEM (S-5200, Hitachi, Tokyo, Japan). The Raman data (HORIBA Xplora plus) of etched silica were collected with a grating of 1200 gr/mm and a laser of 632 nm excitation under ambient conditions.

## 3. Result and Discussion

### 3.1. Amplitude–Frequency and Phase–Frequency Characteristics

The variations of the amplitude–frequency and phase–frequency characteristic curves of the QCRs for three different etching types are given in Figure 2.

The series resonance frequency *f_R_* of QCR is defined by the following equation [20]:(5)$$f_{R}=\frac{1}{2\pi \sqrt{L_{1}C_{1}}}$$
where *L*_1_ is the dynamic equivalent inductance and *C*_1_ is the dynamic equivalent capacitance of QCR. The corresponding parallel resonance frequency *f_A_* is given by the following equation:(6)$$f_{A}=f_{R}\left(1+\frac{C_{1}}{2C_{0}}\right)$$
where *C*_0_ is the static capacitor. The test results in Figure 2a show that after partial etching of the electrodes (type a), the resonance frequency increased to varying degrees. By laser etching, the minimum change frequency could be controlled at 10 Hz, with an accuracy of 1 ppm relative to QRC with a fundamental frequency of 10 MHz, and the corresponding quality factor *Q* decreased by about 6.25%, *C*_1_ increased by 1.2%, and *L*_1_ decreased by 1.2%. As *C*_0_ increased by 1.65%, the corresponding *f_A_* decreased by 3.3‰ (Table 1).

As the mass of the parts of the electrode that were etched away increased, the corresponding frequency change gradually increased. When etching holes did not penetrate the single-sided electrode, the frequency could reach up to 30 KHz, but the corresponding *Q* dropped significantly. The minimum *Q* value was only equivalent to 21.4% of that of an unetched QCR.

The phase–frequency relationship of QCR is shown in the following formula:(7)$$\varphi =\frac{2\pi fL_{1}-\frac{1}{2\pi fC_{1}}}{R_{1}}$$
where *φ* is the phase, *f* is the frequency of the swept signal source, and *R*_1_ is the dynamic equivalent resistance. Clearly, the phase–frequency relationship of QCR will change with *f*. When *f* is smaller than the *f_R_* or larger than the *f_A_*, the phase–frequency characteristics remain essentially unchanged, and when *f* is between the *f_R_* and *f_A_*, the phase fluctuation is small, but obviously lags behind the other two regions. As can be seen from Figure 2a, the phase of the resonator after shallow etching has a spike near its *f_A_*, mainly because the equivalent parameters of the QCR change drastically at this point after etching.

When an electrode on one side of QCR was etched into through-holes (type b), the amplitude–frequency and phase–frequency characteristic curves of the resonator changed significantly, as shown in Figure 2b. The *f_R_* and *f_A_* increased significantly compared to that before etching, *R*_1_ increased significantly from 18 Ω to 508 Ω after etching, and the *Q* was only 3.3% of that of the unetched QCR (Table 1). The phase–frequency characteristic curve was similar to that of Figure 2a; there was also a spike in the phase close to its *f_A_*, and it also lagged behind the other two regions, indicating that the equivalent parameters of the QCR after etching changed drastically, but the phase lag was less than that described in Figure 2a.

However, when the third type (type c) of etching was used, most of the resonators no longer vibrated under external excitation with *R*_1_ sharply increasing from 17 Ω to 855 Ω. The *Q* of the QCR that could resonate occasionally was only 1.6% of that of the unetched resonator (Table 1). As shown in Figure 2c, the phase–frequency curve demonstrated that the phase fluctuation after etching was intensified when the sweep frequency *f* was between the *f_R_* and the *f_A_*. At the same time, a phase near its parallel resonance frequency could be observed.

### 3.2. Morphology Analysis

Figure 3 displays the SEM images of the QCR-related area. It can be seen from the SEM image shown in Figure 3a that the unetched quartz wafer had obvious polishing traces, with small surface undulation and uniform distribution of gullies, but the protrusions of the quartz surface were sharp. From the corresponding cross-sectional view (Figure 3e), the quartz crystal was arranged neatly with a good crystallization condition and perfect orientation consistency. After the quartz wafer (Figure 3b) was directly etched by the laser, the protruding part of the surface was slightly passivated, but due to the short etching time, the change was not obvious. Figure 3f shows a cross-sectional view of the resonator with an as-deposited electrode. It can be seen that the gold was evenly deposited on the surface of the quartz, the surface of the gold film was flat, and its thickness was very uniform. Figure 3c shows the partially etched surface of the gold electrode. Obvious traces of laser work could be seen in the etched area. For the area with a longest etching time, due to the Gaussian distribution of laser energy, the gold electrode at the center had become very thin, and the morphology of the quartz under the ultra-thin gold electrode could be clearly observed. Affected by the anisotropy of thermal conductivity, the radial heat-affected zone was larger than the longitudinal heat-affected zone, while the laser action time was very short, and, thus, the heat-affected zone was small, with a clear etched edge and an etched surface of good quality. Figure 3g shows the cross-sectional view of part of the etched gold electrode. From this perspective, the etching traces of the gold film could be observed more clearly and the laser burning phenomenon was very obvious. As the laser power was Gaussian distributed, the electrode etching depth increased linearly with the number of pulses, while the etching width did not considerably change; thus, it can be surmised that the laser power density was the main factor affecting the etching depth. Due to the higher thermal conductivity of gold than quartz wafers, the actual etching width was larger than the area of the laser spot. This, coupled with the heat-affected area of the laser within a pulse being limited to only a few microns, resulted in the etching pits presenting a wide-top and narrow-bottom shape distribution. However, as shown in Figure 3d, the surface of the quartz wafer after the electrodes (electrodes on both sides) were completely etched by the laser showed a significant difference. First, the protruding part of the surface had obvious passivation, and the protruding part had a tendency to grow upward. The cross-sectional view (Figure 3h) shows that the quartz surface was very flat, and the crystal grain structure in this area had changed locally.

The conductivity of gold is 2.4 × 10^−6^ Ω·cm, and the specific heat capacity is 0.13 kJ/(kg·℃). There are a lot of free electrons in the gold, resulting in a series of physical processes being generated during the laser etching of the resonator electrode. Firstly, photons collided with the free electrons in order to transfer energy, while the laser with a wavelength of 532 nm had a strong projection energy that was consistent with the relationship between wavelength and absorption defined as the Hagen–Rubens relation, which led to a rapid increase in the local temperature of the metal electrode surface. When the temperature reached the melting point, the gold film began to melt or even vaporize. The temperature of the etching area surface was mainly controlled by the evaporation mechanism [21,22]. Due to the good thermal conductivity of gold (315 W/mK), heat was easily transferred to the quartz surface by means of thermal conduction. However, the thermal conductivity of quartz is poor (7.6 W/mK) and the specific heat capacity (0.8 kJ/kg·°C) is larger than that of gold; thus, it was easy to accumulate heat locally, causing the local temperature of quartz to reach and exceed 573 °C. It was extremely easy to cause the quartz to transform from *α*-SiO_2_ to *β*-SiO_2_ and to locally lose piezoelectric properties. This was also verified by the phenomenon of local passivation of the quartz surface in Figure 3d,h. Since the heat conduction penetration depth in the electrode is very shallow under a short period of etching, as long as the electrode is not etched through, the local thermal shock to the surface of the quartz wafer is negligible, and, as such, the resonance equivalent parameters of the QCR do not change much. However, when the cumulative etching time increases, especially when the electrode is etched into through-holes, there are two outcomes. On the one hand, as the electrode area decreases, the energy trapping effect weakens. On the other hand, the local temperature of the quartz wafer will exceed the stable temperature of *α*-SiO_2_ during etching, causing the local vibrational parameters to essentially change, which leads to a strong decay of the resonance performance, or, in extreme cases, results in the QCR no longer resonating. As mentioned above, when the electrodes are fully etched into through-holes, the resonators almost do not oscillate. In such cases, even though the resonance frequency can be measured, it drifts by about 40%.

### 3.3. Raman Spectra

Figure 4 depicted the Raman spectra of quartz wafer surface before and after etching. Six Raman shift peaks at 128 cm^−1^, 206 cm^−1^, 263 cm^−1^, 356 cm^−1^, 394 cm^−1^, 463 cm^−1^ and 805 cm^−1^ can be observed, which are assigned to E(t) + E(l), A_1(broad)_ + E, E(t) + E(l), A_g_ + A_1_, E(t), A_1_ and E(l) vibration modes, respectively [23,24,25,26]. Among them, peaks at 206 cm^−1^ and 463 cm^−1^ are the typical identification Raman shift for *α*-quartz. Different crystal structures have different molecular vibration modes corresponding to different characteristic peaks. Combined with the single-crystal diffraction inset (upper left in Figure 4), the Raman spectra of quartz directly etched by the laser demonstrated no obvious change in structure compared to unetched quartz, since the band gap of quartz is about 8 eV and the laser of 532 nm can completely transmit. However, when a laser was used to etch the gold electrode on the quartz wafer, the Raman spectrum showed some subtle changes, such as a slight increase in the intensity of the peak at 356 cm^−1^ and an increase in the intensity of the Raman peak for *α*-quartz at 206 cm^−1^. Therefore, it can be inferred that the structure of the etched area of the quartz wafer corresponding to the Raman scattering region underwent a slight change.

## 4. Conclusions

Trimming frequency-increasing technology for quartz crystal resonators was achieved by introducing a precisely controllable laser process to develop thin-film thinning technology. In the case of shallow etching of the electrode surface, a clear etched edge of good quality can be obtained, and the degradation in the equivalent parameters of the QCR is very slight. However, the adverse effects brought by excessive etching are also obvious, such as local peeling and warping of the electrode, producing an uneven surface of the quartz wafer and poor uniformity.

Through the analysis of the quartz wafer and electrode material properties, the factors that affect the performance of QCRs are preliminarily explored, and the technical principle and potential applications of laser etching for trimming the frequency of QCRs are confirmed. The experiments in this work show that the laser etching of the gold electrode on the surface of the QCR should be performed using high power, a short pulse width, a small etching aperture, and minimal action time. The experimental results further revealed that the multi-point shallow etching method should be used as much as possible to avoid the occurrence of electrodes being etched through. Finally, the research in this work provides experimental verification for the fine adjustment of QCR frequency by using laser etching technology.

## Figures and Tables

**Figure 1 micromachines-12-00894-f001:**
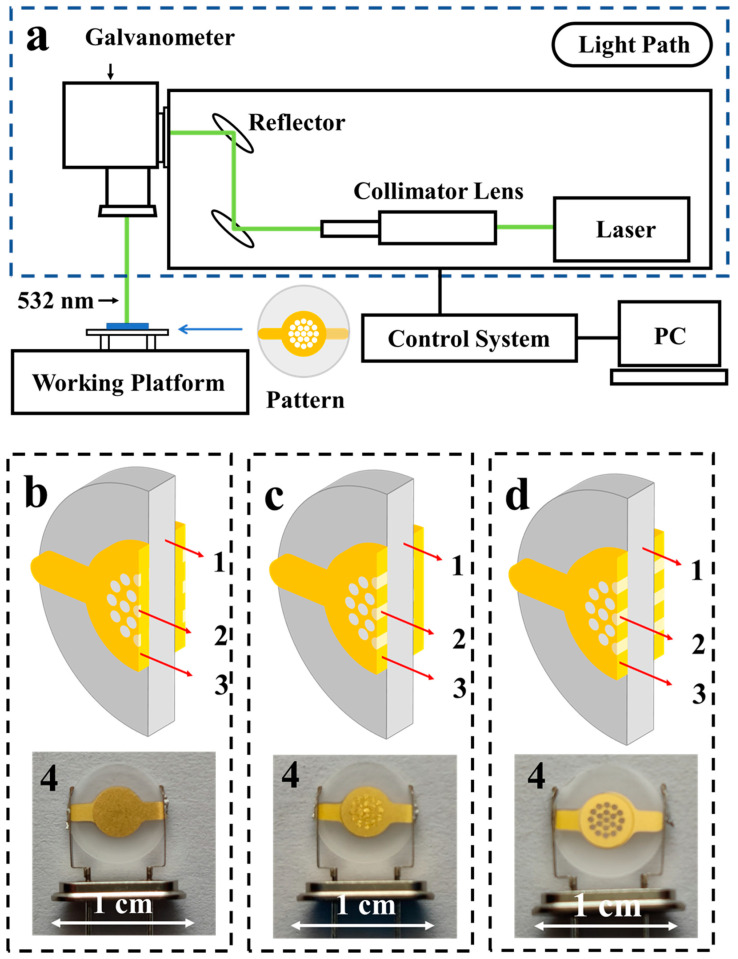
(**a**) Schematic diagram of the laser etching system; (**b**–**d**) show the cross-sectional schematic diagrams of laser etching for QCRs: (**b**) shows that the surface of the electrode was partially etched; (**c**) shows that the electrode on one side of the QCR was etched into through-holes; and (**d**) shows that the electrodes on both sides were etched into through-holes. In these illustrations, 1 is the quartz wafer, 2 is the etched area, 3 is the gold electrode, and 4 is the images of QCRs.

**Figure 2 micromachines-12-00894-f002:**
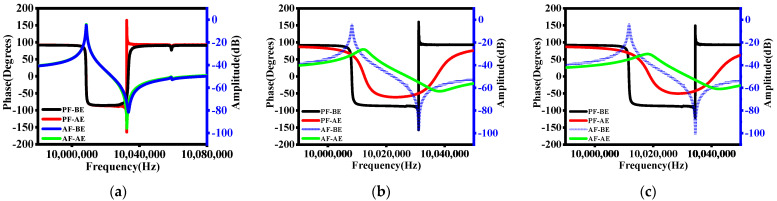
The phase–frequency and amplitude–frequency characteristic curves of the QCRs before and after etching. (**a**) A QCR with electrodes partially etched; (**b**) an electrode on one side of the QCR was etched into through-holes; (**c**) electrodes on both sides were etched into through-holes. AF denotes the amplitude–frequency characteristic curve and PF denotes the phase–frequency characteristic curve. BE represents before etching and AE represents after etching.

**Figure 3 micromachines-12-00894-f003:**
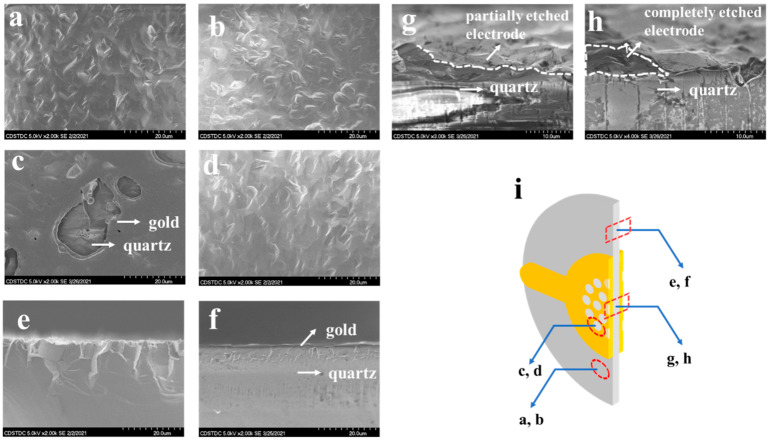
SEM of quartz: (**a**) is the unetched quartz surface; (**b**) is the quartz wafer surface directly etched by a laser; (**c**) is the partially etched surface of a gold electrode; (**d**) is the quartz wafer surface after the laser completely etched the electrodes; (**e**) is the cross-sectional view of the quartz wafer; (**f**) is the cross-sectional view of the resonator with an as-deposited electrode; (**g**) is the cross-sectional view of the resonator with a part of the electrode etched; (**h**) is the cross-sectional view of the resonator with the electrode completely etched; and (**i**) is the schematic diagram of the observed position.

**Figure 4 micromachines-12-00894-f004:**
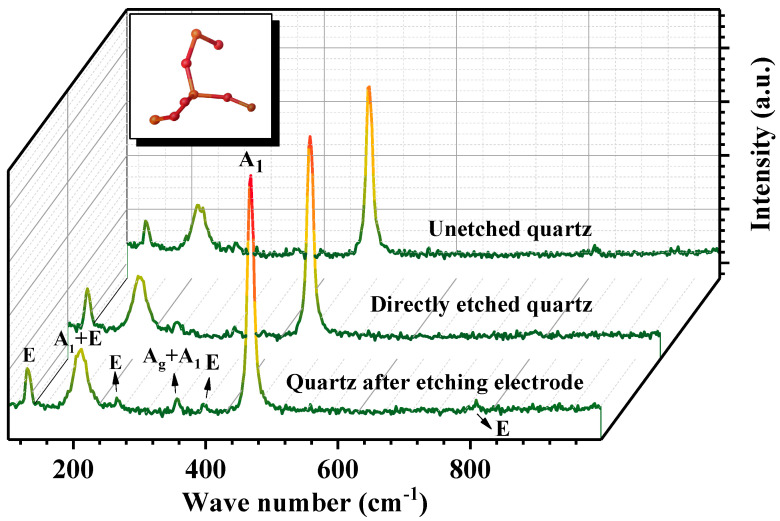
Raman spectra of quartz wafer surface.

**Table 1 micromachines-12-00894-t001:** Equivalent parameters of QCRs.

	Status	F (Hz)	C_0_ (pF)	RR (ohms)	Q (k)	C_1_ (fF)	L (mH)
Type a	Before Etching	10,000,015	5.9237	17.7431	32	28.4628	8.8994
	After Etching	10,000,027	5.8277	18.9616	30	28.1305	9.0045
Type b	Before Etching	10,008,240	3.2577	18.005	60	14.6945	17.2095
	After Etching	10,012,369	3.2095	507.9951	2	14.5876	17.3214
Type c	Before Etching	10,011,762	3.2757	17.358	63	14.57	17.3444
	After Etching	10,018,391	3.1095	855.5109	1	12.8702	19.6091

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
