# Peer review of "Research on Trimming Frequency-Increasing Technology for Quartz Crystal Resonator Using Laser Etching"

_micromachines, 2021, doi:10.3390/mi12080894_

Round 1
Reviewer 1 Report
This paper proposes to use a laser with a wavelength of 532nm to etch the QCR to achieve the precise 10Mhz fundamental frequency of the quartz crystal resonator. The use of laser trimming to adjust the frequency of the quartz wafer is a fairly well-known technology, so this paper lacks innovation.
In addition, I also have the following questions regarding the content of the paper:
- What is the Curie temperature of QCR in this paper? And what is the etching temperature of the 532nm green laser? The temperature and thermal effect of the green laser are usually higher than that of the blue laser. Excessive temperature may affect the piezoelectric properties of quartz.
- Please attach the mode shape and QCR electrode configuration diagram. The frequency drift includes the influence of process and cutting, and it may also be caused by the wrong mode and electrode configuration.
- It is mentioned in Abstract that "quality factor (Q) decreased from 63K to 1K and some resonators have serious frequency drift> 40%", but there is no relevant experimental data in the paper.
- In Fig. 2, why the phase at the resonance frequency is 0 degrees, and why does the anti-resonance disappear?
- In Fig. 3, what is the surface roughness difference of QCR before and after etching? In addition, what is the roughness of the deposited electrode? In (c), (g), (h), different materials should be marked respectively.
Reviewer 2 Report
The manuscript concerns using laser etching as a post-processing method to increase the frequency of quartz crystal resonators. Even though the authors claim this is a more ideal method to trim frequency, the method still needs carefully adjust laser parameters and if the electrode is etched through, the performance of the devices will drop significantly. In addition, the figures are not clear, and the captions are hard to follow. I cannot suggest publication in its current form. Below are my detailed comments:
1. The authors claim the chemically etch is difficult to control the reaction rate, chemical etching tends to lead to non-uniformity in the etching of quartz wafers or electrodes. However, wet etch such as HF is known to be uniform and the rate is well controlled in contrast to what they claim. Can the author comment on this?
2. The method to increase the frequency of the quartz crystal resonators is based on removing materials thus reducing mass. What is the precision of the method? How much did they change the frequency of the resonator?
3. Why can’t they simply CMP the wafer? The rate can be well controlled, and the Q of the devices will remain unaffected. CMP is also easy to access and severs as a standard CMOS process step. What are the advantages of using laser etching then? Can the author comment on this?
4. The authors claim the surface changes of the target material can be ignored during the laser etching process which I hold an opposite opinion. Properties such as thermal conductivity, melting point are significantly different for different materials even if they are inactive materials, such as gold, silver, and platinum. And the parameters of the laser etching process seem to be heavily dependent on the material properties. How could it be ignored then?
5. Figure 1 (a),(b),(c) should be fixed, they are barely visible and there is no scale bar.
6. Figure 2 is very hard to follow, the authors should do a better job in the caption as well as labeling in the figure.
7. Figure 3 (g) and (h) have different scale bar than others. Also, why does (e) looks like crack?
8. In figure 4, it is very hard to see the difference when the lines are on top of each other, I would suggest the authors plot them individually with the same axis, then it will be easier to see the changes. Also, it looks like there is a peak at 800 cm-1, what is that?
Reviewer 3 Report
The authors developed a laser etching process to etch electrodes on a quartz crystal resonator to tune its frequency. The topic is quite interesting and the manuscript looks good in general. I have a few comments here and hope the authors to make some clarifications or changes.
1. In the Introduction section, the overall review of the frequency-tuning techniques is very comprehensive. It may be good to slightly shorten the paragraph.
2. Figure 1: it would be good to make pictures in a, b, c larger. The process is the key in the manuscript so the authors should highlight it.
3. Figure 2: I would recommend to add calculated frequency curves in the plots to compare with the measured curves. The authors have shown the analytical equations so they should make a comparison.
4. Figure 3: it is quite hard to follow those SEM images. Please consider an alternative way to present. One suggestion is to add some illustrations of the quartz crystal resonator and link those SEMs to the structure.
5. Raman spectra: what is the takeaway of the Raman spectroscopy? Are you trying to say there is no change in the quartz wafer material property before and after the laser etching? It seems trivial that the laser etching should not change any material properties. The authors try to explain the change in amplitude, but I could not see how it can be correlated to the laser etching itself. Please clarify.
Round 2
Reviewer 1 Report
I have no other questions.
Reviewer 2 Report
The authors have addressed my previous comments. I am fine with accepting it.